# Urinary Viral Spectrum in Patients with Interstitial Cystitis/Bladder Pain Syndrome and the Clinical Efficacy of Valacyclovir Treatment

**DOI:** 10.3390/biomedicines12030522

**Published:** 2024-02-26

**Authors:** Hann-Chorng Kuo, Chih-Wen Peng, Yuan-Hong Jiang, Jia-Fong Jhang

**Affiliations:** 1Department of Urology, Hualien Tzu Chi Hospital, Buddhist Tzu Chi Medical Foundation, Hualien 640, Taiwan; 2Department of Urology, School of Medicine, Tzu Chi University, Hualien 970, Taiwan; 3Department of Life Science, National Dong Hwa University, Shoufeng 974, Taiwan

**Keywords:** interstitial cystitis, treatment, pathogenesis, etiology, infection

## Abstract

Our previous study showed that the Epstein–Barr virus (EBV) may be the etiology for some patients with interstitial cystitis/bladder pain syndrome (IC/BPS); hence, the current study aimed to investigate the urinary viral spectrum in patients with IC/BPS and the clinical efficacy of valacyclovir. Twenty-eight patients were prospectively enrolled for valacyclovir 500 mg twice a day for 4 weeks. Urine samples were collected from IC/BPS patients and 30 controls. The primary outcome was the difference in the visual analog scale (VAS) pain score, and secondary outcomes included changes in the urinary viral spectrum and urinary inflammatory cytokine level (ClinicalTrials.gov Identifier: NCT05094414). Urinary EBV was detected in 14.2% IC/BPS patients but not in the controls. Urinary John Cunningham virus and BK virus were detected in 18 (64.3%) and 2 (7.1%) patients with IC/BPS, respectively, with similar prevalences noted for the controls. No cytomegalovirus, varicella-zoster virus, or herpes simplex virus was detected in the urine samples. The VAS pain score in patients with IC/BPS significantly decreased after 4 weeks (from 7.5 [5.52–9.0] to 5 [1.5–6.0], *p* = 0.0003). Urinary EBV was undetectable in any sample after valacyclovir treatment, and the decreases in urinary interleukin (IL)-1β (from 0.66 [0.55–0.82] pg/mL to 0.58 [0.55–0.64] pg/mL, *p* = 0.0034), IL-8 (from 6.81 [2.38 to 29.1] pg/mL to 4.33 [1.53–11.04] pg/mL, *p* = 0.0361), IL-10 (from 1.06 [0.94–1.18] pg/mL to 0.92 [0.88–1.02], *p* = 0.0086), and tumor necrosis factor-α (from 1.61 [1.50–1.72] pg/mL to 1.50 [1.44–1.55] pg/mL, *p* = 0.0079) were significant. Valacyclovir could relieve bladder pain, eliminate urinary EBV, and reduce bladder inflammation.

## 1. Introduction

Interstitial cystitis/bladder pain syndrome (IC/BPS) is a common urological disease and is typically characterized by chronic bladder or pelvic pain and frequent urination without bacterial infection [1,2]. In the United States, the estimated prevalence of IC/BPS is 1.08% and 0.66% among women and men, respectively [3]. The histopathology findings of bladders of IC/BPS were characterized by lymphoplasmacytic infiltration, urothelial denudation, and nerve hyperplasia in the bladder [4,5]. Previous studies had used immunochemical staining, polymerase chain reaction (PCR), and next-generation RNA sequencing to reveal the upregulation of inflammatory cytokines, such as interleukin-6 (IL-6), IL-17, and tumor necrosis factor-α (TNF-α), in the bladders of IC/BPS [6,7]. However, the etiology of IC/BPS remains unclear. The current therapeutic strategy for IC/BPS primarily involves reducing bladder inflammation and protecting the urothelium, such as intravesical botulinum toxin injection or hyaluronic acid installation, and studies have revealed significant bladder pain and urinary tract symptoms improvement after treatments [1]. However, symptom relapse is common in patients with IC/BPS, and currently, the curative treatment for IC/BPS is not available.

Although IC/BPS is diagnosed on the basis of the absence of bacterial infection in the bladder [1], the role of viral infection in IC/BPS remains uncertain. Our previous study was the first to provide evidence that Epstein–Barr virus (EBV) infection was present in the bladder of patients with IC/BPS, including IC/BPS patients with Hunner’s lesion (HIC) or without Hunner’s lesion (NHIC). We used in situ hybridization and PCR to assess the presence of EBV, and EBV infection was identified in the bladder specimens of 87.5% and 17.4% of patients with HIC and NHIC, respectively [8]. In a subsequent study, we reported that both latent and lytic EBV infection were present in the bladders of patients with IC/BPS [9], suggesting EBV reactivation in the bladders. We also provided molecular evidence demonstrating that EBV infection is associated with the upregulation of brain-derived neurotrophic factor and EBV-associated inflammatory cytokines in the bladders of patients with IC/BPS [9]. Therefore, infection with EBV may be an etiology in some patients with IC/BPS.

In addition to infection from EBV, infection from polyomaviruses, including John Cunningham virus (JCV) and BK virus (BKV), have been reported to be present in the urine of patients with IC/BPS [10,11], suggesting that polyomavirus infection in the bladder may be associated with IC/BPS. In addition, a report described a patient with IC/BPS who had high levels of urinary polyomavirus and who experienced dramatic symptom relief after intravesical cidofovir treatment [12]. Viral infection in the bladder may be a cause of IC/BPS, and antiviral treatment may thus serve as a curative treatment for patients with IC/BPS. Antiherpesvirus drugs, such as valacyclovir, have been widely used to treat patients with EBV infection because of their strong effects in eliminating herpes virus replication [13,14,15]. As suggested by the results of our previous study that EBV lytic infection in the bladder is associated with IC/BPS [9], valacyclovir may be useful for treating patients with IC/BPS. Hence, our present study first surveyed the types of urinary viruses in patients with IC/BPS. Secondly, we conducted a prospective clinical trial to evaluate the efficacy of oral valacyclovir treatment for patients with IC/BPS, including the effects of IC/BPS symptoms, urinary viruses, and inflammatory cytokines.

## 2. Materials and Methods

### 2.1. Patients and Valacyclovir Treatment Course

This study started in November 2021 and ended in August 2022 (from patient enrollment to laboratory investigations) and was conducted in Hualien Tzu Chi Hospital, Hualien, Taiwan. Patients with IC/BPS were prospectively enrolled between February 2021 and April 2022 in Hualien Tzu Chi Hospital (ClinicalTrials.gov Identifier: NCT05094414). The diagnosis of IC/BPS was made by following the clinical symptom index of the American Urological Association guideline, where IC/BPS is categorized into HIC and NHIC variants on the basis of cystoscopic findings: “An unpleasant sensation (pain, pressure, discomfort) perceived to be related to the urinary bladder, associated with lower urinary tract symptoms of more than six weeks duration, in the absence of infection or other identifiable causes” [1]. Only patients who had IC/BPS symptoms for more than 1 year and experienced symptom recurrence after standard treatment were enrolled in this study. Patients with neurogenic voiding dysfunction, ketamine cystitis, bladder stones, bladder outlet obstruction, a history of any cancer, a history of using any immunosuppressant, previously treated with valacyclovir, or a history of chemotherapy treatment were excluded. Urinary analysis was performed before the valacyclovir treatment, and patients with pyuria (white blood cell count > 5 high power field) were excluded. Urine culture was also investigated to rule out current bacteriuria. A pregnancy test was performed for women of childbearing potential. All patients were refractory to standard treatments for IC/BPS, including oral medications, intravesical botulinum toxin injection, and hyaluronic acid installation. The patients completed a questionnaire that was used to gather data on their baseline clinical features in terms of the visual analog scale (VAS) score for pain (ranged from 0 to 10), quality of life (QoL) score with regard to urinary symptoms, O’Leary–Sant symptom score with the Interstitial Cystitis Symptoms Index (ICSI) score as a component, and the Interstitial Cystitis Problem Index (ICPI). All patients underwent a urodynamic study for cystometric bladder capacity and cystoscopic hydrodistention under general anesthesia for maximal bladder capacity (MBC).

All patients with IC/BPS received open-label oral valacyclovir (Valtrex, GlaxoSmithKline) at a dose of 500 mg twice daily for 4 consecutive weeks. During the period of this clinical trial, the patients were free to take the medications they were already taking but could not take any new medication for IC/BPS. Data on the level of symptom relief and the global response assessment (GRA) score were gathered at the end of weeks 1, 2, and 4 after treatment. The patients also completed the questionnaires, including VAS pain scale, ICPI, ICSI, and QoL, at the end of weeks 1, 2, and 4 after treatment. The GRA scores ranges from 3 to −1, with the scores indicating no symptoms (score: 3), >50% symptom improvement (score: 2), 25% to 50% symptom improvement (score: 1), 0% to 25% symptom improvement (score: 0), and worsened symptoms (score: −1). The patients also completed the questionnaires including VAS pain scale, ICPI, ICSI, QoL at the end of weeks 1, 2, and 4 after treatment. This study was conducted in strict accordance with Good Clinical Practice.

The primary endpoint was the change in VAS pain score from baseline to the end of week 4. Urine samples were obtained again after treatment completion. Patients who visited the urology clinic for anti-incontinence surgery were asked to provide a urine sample as a control for urinary virus investigations. The control patients also had received urine analysis test to exclude asymptomatic urinary tract infection and received urodynamic study to exclude possible bladder diseases such as overactive bladder. The details of the patient enrollment and the protocol of the clinical trial were shown in the Appendix A.

### 2.2. Power and Sample Size

Previous studies revealed the effect size of VAS pain scale in the treatment of IC/BPS were usually medium [16]. We suppose that the valacyclovir treatment of effect was medium and used an effect size of 0.6 to estimate the sample size. Using G*Power 3.1.9.2, we determined that the minimum required sample size was 24 participants at an effect size of 0.6, α value of 0.05, and power (1 − β) of 0.80 for a 2-sided test of the change in VAS pain score from the pretreatment to posttreatment periods. We then enrolled 28 patients with IC/BPS for the treatment, anticipating a dropout rate of no higher than 15%.

### 2.3. Urinary Virus and Inflammatory Cytokine Investigation

We collected 50 mL urine samples at baseline and at 4 weeks after valacyclovir treatment. The samples were placed immediately on ice and then centrifuged at 1800× *g* for 10 min at 4 °C. The supernatant was preserved in a freezer at −80 °C. Before further analysis was completed, the frozen urine samples were centrifuged at 12,000× *g* for 15 min at 4 °C, and the supernatants were used for subsequent experiments. The centrifuged supernatants (1 mL) were sent to the Medical Laboratory Department in Hualien Tzu Chi Hospital for urinary virus investigation. The virus DNA polymerase chain reaction (PCR) protocol was developed in our laboratory department and passed the viral load survey offered by the College of American Pathologists. The urinary DNA extraction used the LabTurbo 48 compact system (LabTurbo, Taipei, Taiwan) with Virus Mini Kit (Catalog No: LVN480-1000; LabTurbo, Taipei, Taiwan). First, proteinase K was added to the urine samples. Lysis of the sample was performed by adding lysis buffer VLL and incubating for 15 min at 57 °C. We moved the sample to commercial column and then removed the extra buffer with ethanol and suction. The sample was washed with wash buffer LW1 and suctioned twice. Finally, elution buffer CCEB was added to elute the DNA/RNA extract, which was then ready for real-time PCR. The real-time PCR was performed with Rotor-Gene Q 5plex HRM (QIAGEN, Hilden, German) and the TaqMan system. The qPCR detection protocol was as follows: (1) The reaction mix in each well included oasig 10 μL, primer/probe mix 1 μL, and RNase/DNase free water 4 μL. Then, 5 μL of DNA template was added into each reaction mix. The standard followed the manufacturer’s instructions. (2) qPCR amplification took 50 cycles as follows: enzyme activation at 95 °C for 2 min, denaturation at 95 °C for 10 s, and data collection at 60 °C for 1 min. The EBV DNA detection was targeted on a nonglycosylated membrane protein (BNRF1) gene with a commercialized kit (Primerdesign, Chandler’s Ford, UK). The BKV and JCV DNA detection was also performed with commercialized kits from Primerdesign. The BKV detection was targeted on a non-coding region, and the primers had 100% homology with more than 95% of reference sequences in the National Center for Biotechnology Information database; therefore, the quantification profile was very broad. The urinary VZV and HSV were investigated with another commercialized kit (LightMix, Roche, Basel, Switzerland, Cat.-No. 40-0358-96 and Cat.-No. 40-0562-32, respectively). We captured the results using a reader.

The level of urinary inflammatory cytokines was measured using a commercial Milliplex human cytokine/chemokine magnetic bead-based panel kit (Millipore, Darmstadt, Germany); these cytokines were IL-1β, IL-6, IL-8, IL-10, brain-derived neurotrophic factor (BDNF), tumor necrosis factor alpha (TNF-α), monocyte chemoattractant protein-1 (MCP-1), and macrophage inflammatory protein-1 alpha (MIP-1a). The procedure was identical to that in our previous study [17]. The analytes were measured by using the multiplex kit (catalog number: HCYTMAG-60K-PX30). A total of 25 μL assay buffer, 25 μL urine sample, and 25 μL beads were sequentially added to 96-well plates (panel kits), and the plates were incubated overnight in the dark at 4 °C. The contents of the wells were removed, and the plates were washed twice with 200 μL wash buffer. Then, 25 μL of detection antibody were added to each well, and the plates were incubated in the dark on a shaker plate for 1 h at room temperature. Next, 25 μL of streptavidin/phycoerythrin solution was added into each well (to form a capture sandwich immunoassay); incubation was then performed in the dark for 30 min at room temperature. The well contents were again removed, and the plates were washed twice with 200 μL wash buffer. Finally, 150 μL of sheath fluid was added, and the plates were evaluated on the MAGPIX instrument with xPONENT software (version 4.2). Median fluorescence intensities of all cytokine/chemokine targets were analyzed to calculate the corresponding cytokine/chemokine concentrations in the urine samples. The details of the laboratory procedures to detect urinary virus and the detection power of the used kits were showed in the section of Appendix A.

### 2.4. Study Approval

This study was approved by the Institutional Review Board and Ethics Committee of Hualien Tzu Chi Hospital (IRB number: 108-45-A). All patients were informed about this study’s rationale and procedures and the risks of participation. All parts of this study comply with the Declaration of Helsinki.

### 2.5. Statistical Analysis

The clinical and laboratory results for each group were compared using nonparametric methods, namely the Mann–Whitney U test for 2 nonpaired groups and the Wilcoxon matched-pair signed-rank test for 2 paired groups. The urinary viral spectrum between IC/BPS patients and the controls were compared with the Chi-square test. Significance was indicated by *p* < 0.05. All statistical analyses were performed in GraphPad Prism 8 (Boston, MA, USA).

## 3. Results

### 3.1. Urinary Viral Profile of Patients with IC/BPS

Among the 28 patients with IC/BPS (mean age: 55.3 ± 13.4 years), 4 had HIC and 24 had NHIC. No patient dropped out of this study, and the data of all 28 patients were included in the final analysis. All patients completed the 4-week treatment course and provided a urine sample at baseline and at 4 weeks of valacyclovir treatment. All patients used pain control medications (acetaminophen or celecoxib), and the baseline mean VAS pain scale among these patients was 6.79 ± 2.67 (median: 7.5, 25th–75th percentile: 5.25–9.0). Data on the other baseline characteristics are presented in Table 1. Among the 28 patients, 4 (all of whom had HIC) had detectable EBV in urine (14.2%, Figure 1), 2 had detectable BKV in urine (7.1%), and 18 had detectable JCV in urine (64.3%). Among the urine samples of the 30 controls, 15 had detectable JCV, and 1 had detectable BKV; none had detectable EBV. The ratios of detectable urinary JCV and BKV did not significantly differ between the patients with IC/BPS and the controls (JCV: 64.3% vs. 50.0%, *p* = 0.436; BKV: 7.1% vs. 3.3%. *p* = 0.951). Urinary HSV, VZV, and CMV were not detected for either the patients or the controls. The patients with presence of urinary EBV had significantly higher VAS bladder pain score and smaller MBC than the patients without urinary EBV (Table 2). The presence of urinary JCV was associated with a smaller MBC but not associated with bladder pain or ICSI (Table 2). Patients with any positive urinary virus had significantly smaller CBC and MBC than those without any positive urinary virus (Table 2).

### 3.2. Valacyclovir for Pain Relief in Patients with IC/BPS

The VAS pain score of the patients with IC/BPS improved significantly after 1 week of medication, and the level of pain progressively decreased by the end of the first month (Figure 2A, from 6.78 ± 2.67 [7.5, 5.52–9.0] at baseline to 5.10 ± 3.29 [6, 1.0–7.75] at first week, 4.82 ± 3.16 [6, 0.75–7.0] at second week, and 4.46 ± 2.95 [5, 1.5–6.0] at fourth week; *p* = 0.0003, 0.0002, and 0.0003, respectively. Data are presented as mean ± standard deviation [median, 25th–75th percentile]). The decrease in mean VAS score at the end of the first month was 2.32 ± 3.27. The QoL index also exhibited significant improvement at 4 weeks after treatment (Figure 2A, from 4.46 ± 1.18 to 3.68 ± 1.26, *p* = 0.0062). No improvement in ICPI or ICSI was noted among the patients at the end of this study. The mean GRA at the end of the study was 1.07 ± 1.36. Of the 28 patients, 7 (25%) had a GRA score of 3 (symptom free), and 6 (21.4%) experienced no pain (Figure 2A). Nine (32.1%) of the IC/BPS patients had GRA score more than 2 (symptoms improved more than 50%). In contrast, eight patients (28.5%) exhibited no improvement in symptoms (GRA = 0), and two patients had worse IC/BPS symptoms (GRA = −1).

Ten patients (35.7%) were classified as responders after 1 month of treatment, with response defined by a decrease of ≥3 in VAS pain score after valacyclovir treatment. The responders had a higher baseline VAS pain scale score (8.40 ± 2.01 vs. 5.89 ± 2.60, *p* = 0.0102), grade of glomerulation hemorrhage (2.7 ± 1.3 vs. 1.7 ± 0.6, *p* = 0.0415), and MBC (545.0 ± 249.9 mL vs. 781.1 ± 189.5 mL, *p* = 0.0206, Figure 2B) than the nonresponders did. All four patients with HIC were responders, and two were free of pain after treatment completion. The decrease in VAS pain score was also significant among the patients with NHIC (decreasing from 6.38 ± 2.65 to 4.79 ± 2.93, *p* = 0.011). No patients with urinary EBV at baseline had detectable EBV in their urine after treatment (Figure 1). The percentage of patients who had urinary JCV did not significantly differ before (64.3%) versus after (57.1%) valacyclovir treatment (*p* = 0.687). With regard to adverse effects, three patients reported mild dizziness after valacyclovir treatment, and one of these three patients also had mild general weakness. All three patients had symptom relief without the use of additional medication, and no patient discontinued the valacyclovir treatment because of adverse effects.

### 3.3. Changes in Urinary EBV and Inflammatory Cytokine Levels from before to after Valacyclovir Treatment

In this study, we investigated the levels of urinary inflammatory cytokines in the patients with IC/BPS, and the results revealed significantly decreased levels of urinary IL-1β, IL-8, IL-10, and TNF-α after 4 weeks of valacyclovir treatment (Figure 3). In contrast, the level of urinary IL-6, BDNF, MIP-1α, or MCP-1 in the patients with IC/BPS did not have significantly change after the treatment. The responders in this study exhibited significant decreases in the levels of IL-1β, IL-8, IL-10, and BDNF (Figure 4) after the valacyclovir treatment, while the nonresponders exhibited no significant change in the level of any cytokine. The responders also had higher levels of baseline urinary IL-1β than the nonresponders (Figure 4, *p* = 0.009).

## 4. Discussion

In the present study, we investigated the urinary viral spectrum of patients with IC/BPS, and urinary EBV was detected only in the patients with HIC. This is the first clinical trial to investigate the efficacy of antiviral medication for patients with IC/BPS, and the results indicate that valacyclovir treatment can reduce bladder pain in patients with IC/BPS regardless of whether they have HIC. The levels of urinary inflammatory cytokines in the patients with IC/BPS significantly decreased after valacyclovir treatment, and urinary EBV was also eliminated in the patients with HIC.

In addition to urinary bacterial infection, researchers have been investigating the role of viral infection in IC/BPS since the 1970s [18]. The results obtained using various laboratory techniques have indicated the presence of CMV, adenovirus, BKV, JCV, HSV, and VZV in the urine or bladder tissues of patients with IC/BPS [10,11,18,19,20,21]. In 2020, the Multidisciplinary Approach to the Study of Chronic Pelvic Pain Research Network used next-generation sequencing to analyze urine samples from patients with IC/BPS and detected two human polyomaviruses—BKV and JCV [11]. The present study used PCR to investigate six kinds of viruses and also detected EBV, JCV, and BKV in the urine samples of patients with IC/BPS (Figure 1); however, urinary CMV, VZV, and HSV were not detected. Although urinary JCV could be detected in 64.3% of patients with IC/BPS (Figure 1), it had a similar prevalence between the patients and controls. In contrast, EBV could be detected in all patients with HIC but not in the controls or patients with NHIC. A previous study reported that urinary EBV could be detected in 93% patients with infectious mononucleosis [22], suggesting that detectable urinary EBV is associated with EBV lytic infection. The detectable urinary EBV in the patients with HIC also supported our previous finding of the coexistence of EBV persistence and reactivation in the bladders of patients with HIC [9]. In our previous study, we reported a higher proportion of detectable EBV in the IC/BPS bladders, and the current study showed a relatively lower proportion of detectable EBV in the urine samples from patients with IC/BPS. IC/BPS patients with EBV in their bladders may not also have detectable urinary EBV. Bladder viral infection might be a possible etiology in some patients with IC/BPS, but prevalence should be investigated with the studies which enrolled more patients.

In a clinical trial, valacyclovir that was administered orally at a dose of 1000 mg every 8 h over 28 days was effective for treating patients with EBV-reactivation-associated oral hairy leukoplakia [23]. Our results indicate the presence of EBV lytic infection in the bladder in IC/BPS [9]; thus, valacyclovir may be used to treat IC/BPS. Although only four patients had urinary EBV in this study (Figure 1), EBV undetectable in the urine specimens did not mean that the bladders specimens were free for EBV infection. The IC/BPS patients might still have EBV latency infection in their blader, but their urine samples did not have EBV. Theoretically, we should only use valacyclovir to treat IC/BPS patients with EBV, but we could not identify EBV infection only using urine test. Hence, we also included the patients without urinary EBV to evaluate the effect of valacyclovir treatment. In our clinical trial, valacyclovir significantly reduced the level of bladder pain in the patients with IC/BPS (Figure 2A), with greater pain relief noted among the patients with HIC and the patients with more severe baseline bladder pain (Figure 2B). Although bladder pain relief was also observed in the patients without urinary EBV, the bladder pain reduction in the patients was only 24.9%, and the response rate was only 29.4%. The improvement in patients without urinary EBV might be just placebo effects. No patients with baseline urinary EBV had urinary EBV after valacyclovir treatment (Figure 1), suggesting oral valacyclovir could inhibit EBV replication in the bladders. The decline in the levels of EBV-associated inflammatory cytokines in urine also suggested improvement in the EBV infection and inflammation in the bladders of the patients with IC/BPS (Figure 3), and only the responders to valacyclovir treatment had decreased urinary inflammatory cytokine levels (Figure 4). In addition, the treatment responders had higher levels of baseline urinary inflammatory cytokines, specifically IL-1β (Figure 4), indicating that urinary inflammatory cytokines may serve as a marker of response to valacyclovir treatment. However, current antiherpesvirus drugs do not help in eliminating latent EBV [24]. Persistent EBV latency in bladders may cause chronic urinary symptoms in patients with IC/BPS who receive valacyclovir. Although six patients reported no pain at the end of this study, their symptoms may relapse because EBV latency was not eradicated. When the bladder pain relapse, the EBV-associated IC/BPS patients may need to receive valacyclovir treatment again. All of the anti-herpes medications, including acyclovir, valacyclovir and famciclovir, were effective to inhibit EBV replication and theoretically could be used to treat IC/BPS with EBV infection. Novel EBNA1-targeted inhibitors have been developed for treating EBV latent infection [25] and may be effective against IC/BPS. Oral valacyclovir might cause adverse effect such as nausea, vomiting, headaches, dizziness, diarrhea, or constipation [26]. For the IC/BPS patients without urinary EBV, clinicians should consider the benefit and potential adverse effect of the valacyclovir treatment.

This study’s limitations are its small sample size and lack of a placebo control group, which can be addressed in future studies. The improvement of pain in patients has a high potential to be affected by the placebo effect. Further multi-institutional placebo-controlled randomized clinical trial is necessary to prove the effects of valacyclovir on IC/BPS patients with urinary EBV. In the control group, only women were enrolled for urine sample, but the study group included four male patients. Sexual differences might have impacts on urinary viral spectrum. Using the VAS pain scale of the pain may not be an accurate method to quantify pain in the patients because they needed to select from a finite number of pain description and may not find some that appropriately reflects their discomfort. Compared with a previous study of valacyclovir for oral hairy leukoplakia [23], this preliminary study used a low dose of valacyclovir, 500 mg twice daily. Future studies should determine the optimal dose and duration of valacyclovir treatment for IC/BPS. Our findings serve as evidence that EBV infection in bladders with HIC plays a role in the pathophysiology of IC/BPS [9]; however, the pathological contribution of EBV to NHIC IC/BPS remains unclear. Nonetheless, a possible link between the herpesvirus (and EBV) and the pathogenesis of NHIC remains to be uncovered, as indicated by the current study’s observations of bladder pain relief in the patients with NHIC who received valacyclovir. Future studies that aim to develop antiEBV protocols are likely to deliver new methods for curing IC/BPS.

## 5. Conclusions

Valacyclovir treatment may significantly relieve bladder pain in patients with IC/BPS, especially those with more severe baseline symptoms. Urinary EBV could be identified in the patients with HIC and was eliminated after 4 weeks of valacyclovir treatment. The levels of urinary inflammatory cytokines were also significantly decreased after valacyclovir treatment.

## Figures and Tables

**Figure 1 biomedicines-12-00522-f001:**
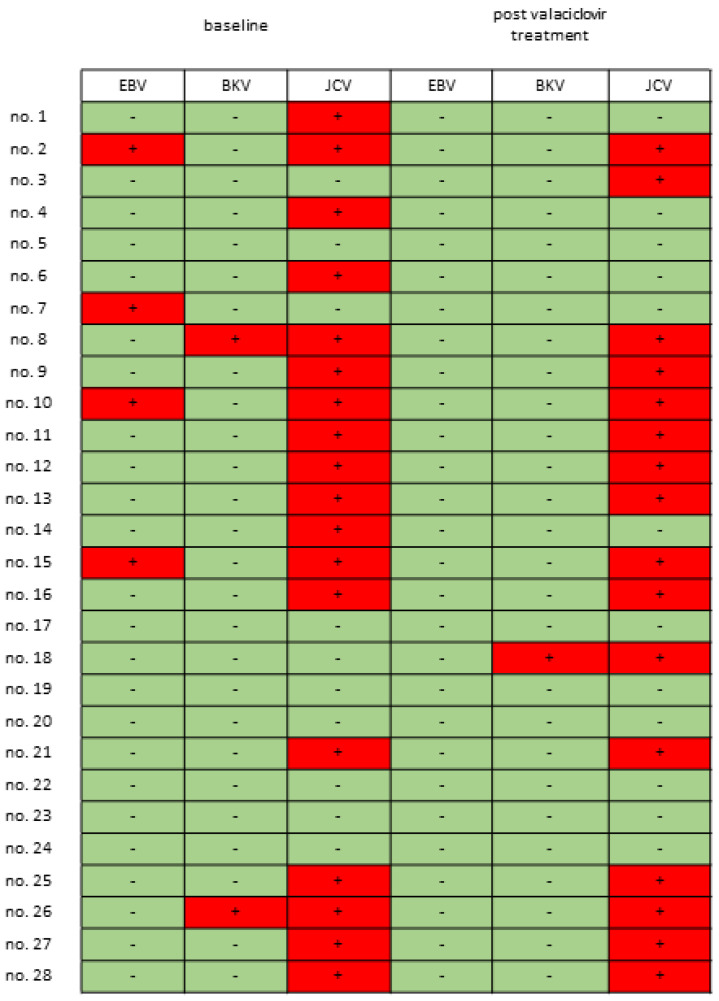
Urinary viral spectrum in patients with interstitial cystitis/bladder pain syndrome (IC/BPS) before and after valacyclovir treatment. The prevalence rates of urinary John Cunningham virus (JCV) and BK virus (BKV) among the patients with IC/BPS did not significantly differ between the baseline and posttreatment periods.

**Figure 2 biomedicines-12-00522-f002:**
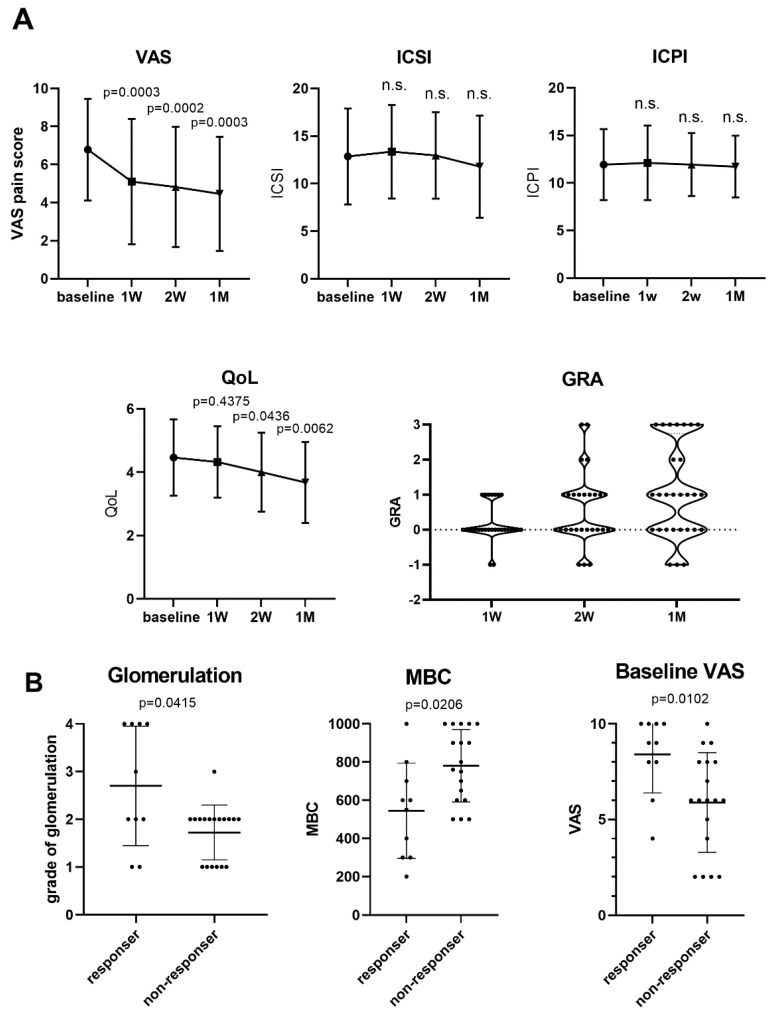
Clinical efficacy of valacyclovir treatment for patients with interstitial cystitis/bladder pain syndrome (IC/BPS). (**A**) Changes in symptom scores in 28 patients with IC/BPS who received valacyclovir. The visual analog scale (VAS) pain score significantly decreased after the first week of treatment. The Interstitial Cystitis Symptoms Index (ICSI) and Interstitial Cystitis Problem Index (ICPI) exhibited no significant change (*p* > 0.05 in Wilcoxon matched-pair signed-rank test relative to baseline). The quality of life (QoL) score significantly improved after the second week of treatment. After 1 month of treatment, 9 patients had a global response assessment (GRA) ≥ 2. (**B**) The responders to treatment had a higher baseline glomerulation grade, VAS pain score, and lower maximal bladder capacity (MBC).

**Figure 3 biomedicines-12-00522-f003:**
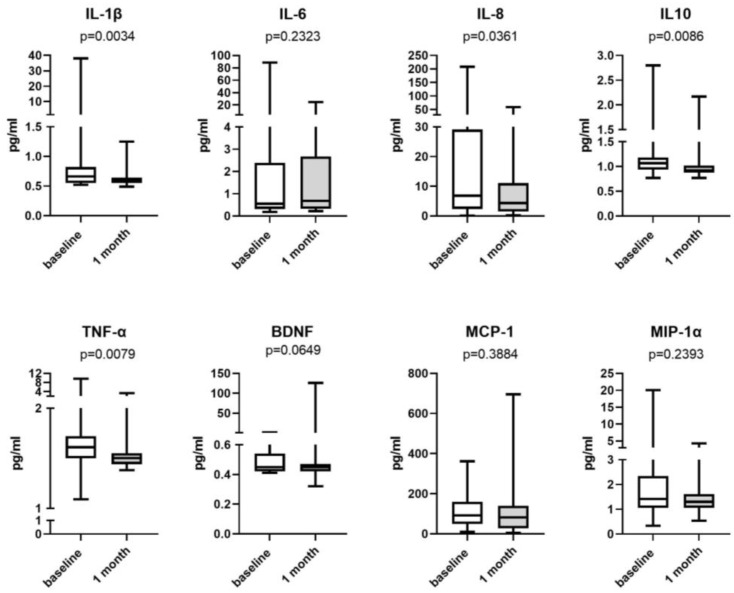
Urinary cytokine levels in patients with interstitial cystitis/bladder pain syndrome before and after valacyclovir treatment. Urinary interleukin (IL)-1β, IL-8, IL-10, and tumor necrosis factor (TNF)-α levels significantly decreased after 1 month of treatment. BDNF = brain-derived neurotrophic factor, MCP-1 = monocyte chemoattractant protein-1, MIP-1a = macrophage inflammatory protein-1 alpha.

**Figure 4 biomedicines-12-00522-f004:**
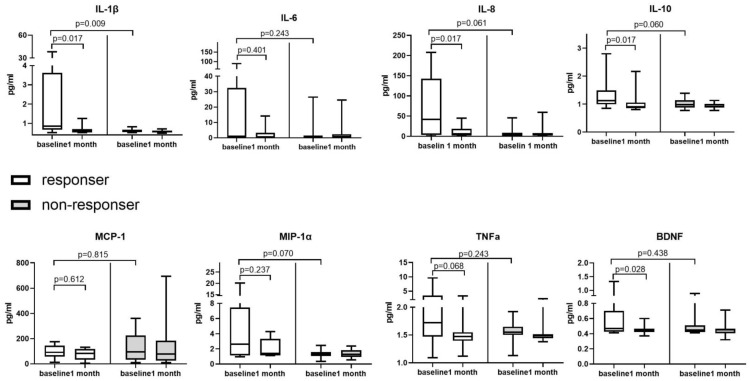
Urinary cytokine levels in valacyclovir treatment responders and nonresponders. Urinary interleukin (IL)-1β, IL-8, IL-10 and brain-derived neurotrophic factor (BDNF) levels significantly decreased in the responders but not the nonresponders. The responders had higher levels of baseline urinary IL-1β than the nonresponders. MCP-1 = monocyte chemoattractant protein-1, MIP-1a = macrophage inflammatory protein-1 alpha.

**Table 1 biomedicines-12-00522-t001:** Baseline characteristics of the IC/BPS patients who received the valacyclovir treatment and the control subjects.

	Baseline Characteristics	
	Patients with IC/BPS (*n* = 28)	Control Subjects (*n* = 30)
Age	55.3 ± 13.4	54.7 ± 5.70
Sex	24 female/4 male	All female
Hunner’s lesion	4 HIC/24 NHIC	N/A
VAS	6.79 ± 2.67 (7.5, [5.25–9.0])	N/A
ICSI	12.9 ± 5.04 (13, [11.25–16.5])	N/A
ICPI	11.9 ± 3.73 (12, [11–13.75])	N/A
CBC (mL)	240.2 ± 116.7 (239.5. [145.25–329.75])	N/A
QoL	4.46 ± 1.20 (4. [3–6])	N/A
MBC (mL)	664.6 ± 255.8 (700, [512.5–900])	N/A
Grade of glomerulation hemorrhage	Gr. 1: 8; Gr. 2: 14; Gr. 3: 2; Gr. 4: 4	N/A

Values are presented as mean ± standard deviation (median, [25th–75th percentile]). VAS: visual analog scale score for pain; ICSI and ICPI: interstitial cystitis symptom indexes and problem indexes; CBC: cystometric bladder capacity; QoL: quality of life due to urinary symptoms; MBC: maximal bladder capacity under cystoscopic hydrodistention. N/A: Not applicable.

**Table 2 biomedicines-12-00522-t002:** The baseline clinical parameters in the IC/BPS patients with or without urinary virus.

	EBV Positive *n* = 4	EBV Negative *n* = 24	*p*-Value	JCV Positive *n* = 18	JCV Negative *n* = 10	*p*-Value	Any Virus Positive *n* = 19	Any Virus Negative *n* = 9	*p*-Value
**ICSI**	19.5	12.5	0.128	13	13	0.384	13	12	0.758
	[9.25, 20.0]	[11.3, 15.0]		[9.0, 15.5]	[12.0, 20.0]		[10.3, 17.0]	[12.0, 14.8]	
**ICPI**	16	12	0.074	12	12	0.640	12	12	0.434
	[11.5, 19.0]	[11.0, 13.0]		[10.8, 13.])	[10.8, 16.0]		[11.3, 14.8]	[10.3, 12.8]	
**VAS**	9.5	6	0.031	6	8	0.308	7	7.5	0.918
	[8.25, 10.0]	[4.25, 8.75]		[3.50, 9.0]	[5.75, 9.25]		[4.5, 9.75]	[5.25, 8.75]	
**CBC** **(mL)**	124.5	188.5	0.131	162	231	0.314	153	298	0.042
	[68.5, 167]	[110, 307)		[103, 247]	[91.0, 340]		[91.8, 241]	[181, 341]	
**MBC** **(mL)**	400	700	0.007	575	900	0.005	575	900	0.002
	[225, 500]	[563, 790)		[500, 713]	[675, 1000]		[500, 738]	[713, 1000]	

Data are presented as median [IQR, 25th–75th percentile]. The *p*-value was analyzed with Mann–Whitney U test. VAS: visual analog scale score for pain; ICSI and ICPI: interstitial cystitis symptom indexes and problem indexes; CBC: cystometric bladder capacity; MBC: maximal bladder capacity under cystoscopic hydrodistention.

## Data Availability

Data are available from the corresponding author upon reasonable request.

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
