# Peer review of "Urinary Viral Spectrum in Patients with Interstitial Cystitis/Bladder Pain Syndrome and the Clinical Efficacy of Valacyclovir Treatment"

_biomedicines, 2024, doi:10.3390/biomedicines12030522_

Round 1

Reviewer 1 Report

Comments and Suggestions for Authors

The authors present a prospective clinical trial on the effect of valacyclovir on interstitial cystitis (IC). They found that 4 patients (14%) had urinary EBV at baseline, and none had urinary EBV after treatment. They report a significant improvement in symptoms 4 weeks after a course of oral valacyclovir. They conclude that EBV plays a role in IC and that valacyclovir could have a role in management. The study is novel and adds value to the literature. The manuscript might be improved by considering the following comments and questions.

As the authors state in the discussion, the small sample size and absence of a control group severely limits the results. The primary outcome was improvement in pain, which has high potential to be affected by placebo.

The overall improvement in pain from 6.8 to 4.5 was moderate. All 4 of the patients with baseline urinary EBV had a significant improvement in pain (GRA > 2), but only 3 of the 24 patients (12.5%) without baseline EBV had a significant improvement in pain. 12.5% significant response rate in pain is similar to what would be expected from a placebo treatment. Therefore, I would only really pursue valacyclovir as a treatment for IC patients with baseline EBV, especially since there is no clear mechanism to explain the improvement in patients without EBV. Most urology clinics do not currently have a practical way of testing for urinary EBV.

I encourage the authors to conduct a larger multi-institutional placebo-controlled randomized clinical trial on the effects of valacyclovir on IC patients w urinary EBV. If such a trial demonstrated efficacy and we had a practical and cost-effective way of detecting urinary EBV in urology clinics, this could change the IC guidelines to include a test for EBV. In fact, the presence of urinary EBV might indicate a new diagnosis (viral cystitis) that is distinct from IC.

Author Response

The authors present a prospective clinical trial on the effect of valacyclovir on interstitial cystitis (IC). They found that 4 patients (14%) had urinary EBV at baseline, and none had urinary EBV after treatment. They report a significant improvement in symptoms 4 weeks after a course of oral valacyclovir. They conclude that EBV plays a role in IC and that valacyclovir could have a role in management. The study is novel and adds value to the literature. The manuscript might be improved by considering the following comments and questions.

As the authors state in the discussion, the small sample size and absence of a control group severely limits the results. The primary outcome was improvement in pain, which has high potential to be affected by placebo.

Reply: Thanks for your comment. Indeed, the improvement of pain in the patients has high potential to be affected by placebo effect. We make a notice in the limitation section (page 12, line 26 to 27).

The overall improvement in pain from 6.8 to 4.5 was moderate. All 4 of the patients with baseline urinary EBV had a significant improvement in pain (GRA > 2), but only 3 of the 24 patients (12.5%) without baseline EBV had a significant improvement in pain. 12.5% significant response rate in pain is similar to what would be expected from a placebo treatment. Therefore, I would only really pursue valacyclovir as a treatment for IC patients with baseline EBV, especially since there is no clear mechanism to explain the improvement in patients without EBV. Most urology clinics do not currently have a practical way of testing for urinary EBV.

Reply: Thanks for your comment. For patients without baseline urinary EBV (n=24), 4 patients had GRA =3 and 3 patients had GRA =2. Since the definition of responder in study was GRA ≥2, the response rate in patients without urinary EBV was 7/24 (29.1%). The reduction of mean VAS pain score in IC/BPS patients without urinary EBV was only 24.9% (from 6.38 to 4.79). Although the reduction of VAS pain in IC/BPS patients without urinary EBV is statistically significant, the low proportion of pain score improvement and response rate suggestion the effect might be just placebo effect. We added some notification in the discussion section (page 12, line 1 to 4).

I encourage the authors to conduct a larger multi-institutional placebo-controlled randomized clinical trial on the effects of valacyclovir on IC patients w urinary EBV. If such a trial demonstrated efficacy and we had a practical and cost-effective way of detecting urinary EBV in urology clinics, this could change the IC guidelines to include a test for EBV. In fact, the presence of urinary EBV might indicate a new diagnosis (viral cystitis) that is distinct from IC.

Reply: Thanks for your kindly encouragement. Further multi-institutional placebo-controlled randomized clinical trial is necessary to prove the effects of valacyclovir on IC/BPS patients with urinary EBV (page 12, line 27 to 29). EBV detection in blood is common laboratory tool to diagnosis and evaluate the patients with lymphoma, and most medical center could use PCR to detect EBV in blood. The procedures to do PCR in urine and blood sample to detect EBV are identically after the centrifugation. In fact, in the medical laboratory department of our hospital also used their routine kit to detect EBV in urine samples from the patients with IC/BPS.

Reviewer 2 Report

Comments and Suggestions for Authors

This is a unique report of a successful antiviral drug for interstitial cystitis.

Although the number of cases is small, it is interesting to note that the effect was remarkable in EBV-positive cases.

Please let us know if you have a case that shows how long the effect lasted after treatment.

You mention that it may be necessary to re-administer the drug eventually, but will re-administering the drug still be successful?

Is there any possibility of antivirals other than valacyclovir?

Author Response

This is a unique report of a successful antiviral drug for interstitial cystitis.

Although the number of cases is small, it is interesting to note that the effect was remarkable in EBV-positive cases.

Please let us know if you have a case that shows how long the effect lasted after treatment.

Reply: Thanks for your comment. In this study, the patients only used valacyclovir for 1 month. Although the following treatment outcomes were not well followed after the clinical trial was completed, most responders of this study asked to keep valacyclovir treatment for 3 to 6 months and had stable bladder symptoms. We noticed that most patients may have symptoms relapse in 3 months after termination of the valacyclovir treatment. We do not add the above data to the manuscript because the patients were not well followed after the trial. 

You mention that it may be necessary to re-administer the drug eventually, but will re-administering the drug still be successful?

Reply: Thanks for your comment. Seven of the responder patients received the valacyclovir treatment after symptoms relapse, and five of them had subjectively symptoms improved. We do not add the above data to the manuscript because the treatment outcome was not well followed.

Is there any possibility of antivirals other than valacyclovir?

Reply: Thanks for your comment. For treating EBV, currently there is no specific medication and most patients with EBV infection were treated with anti-herpes medications. In our country, there are 3 most common oral anti-herpes medications: acyclovir, valacyclovir and famciclovir. All of the anti-herpes medications were effective to inhibit EBV replication and theoretically could be used to treat IC/BPS with EBV infection. Recently, novel EBNA1-targeted inhibitors have been developed for treating EBV latent infection and may be effective against IC/BPS. We added the above discussion into the manuscript. (page 12, line 17 to 21)

Reviewer 3 Report

Comments and Suggestions for Authors

General comments

- The main concern is related to the applicability; you did not succeed to describe the methods in sufficient detail to allow the reproduction of the study.

- A change in VAS scale of 0 to 10 equal to 0.6 is without clinical relevance so your study is worthless. How do you quantify a change of 0.6 (the mean is not the appropriate centrality measure for such a small sample)?

- You previously reported Epstein-Barr virus 87.5% and 17.4% of patients with HIC and NHIC but your current results are "Of the 28 patients with IC/BPS, 4 had urinary 201 EBV at baseline and none had urinary EBV after treatment." If your hypothesis was that patients had EBV and you treated EBV with valacyclovir 4 subjects, it is a case series, not an original article. Your inclusion criteria were wrong; for such hypothesis, you must include only patients with EBV. 

- You cited up to 10 articles of co-authors out of 30 references. This is too much.

Abstract

- Lines 11-13 incorporate two sentences.

- When and where the study was conducted?

- "valacyclovir 500 mg twice a day for 4 weeks" this treatment is current practice or not?

- "urinary inflammatory cytokine level" which cytochine?

- "The VAS pain score in patients with IC/BPS significantly decreased after 4 weeks of valacyclovir" not specific. Decrease with how many points? Which was the initial VAS and the 4 weeks VAS?

- "the levels of urinary interleukin (IL)-1β, IL-8, IL-10, and tumor necrosis factor-α were decreased." unspecific.

- "Valacyclovir could relieve bladder pain, eliminate urinary EBV, and reduce bladder inflammation." this sentence was also true before your study. It is expected to have here the conclusion.

Introduction

- Please avoid repetition of "bladder" in the same sentence.

- Please do not use expressions such as "A recent study".

- "Previous studies also revealed upregulation of inflammatory cytokines, such as interleukin-6 (IL-6), IL-17, and tumor necrosis factor-α (TNF-α) in the bladders of IC/BPS.[6,7]" Please be specific.

- "The current therapeutic strategy for IC/BPS primarily involves reducing bladder inflammation and protecting the urothelium, such as intravesical botulinum toxin injection or hyaluronic acid installation [1]."  Which is the efficacy?

- "Our previous study was the first to provide evidence that Epstein-Barr virus (EBV) infection was present in the bladder of patients with IC/BPS" how frequent?

- "In situ" should be read as "in situ".

- Write the aim of your study at past tense.

Materials and Methods

- "2021 and 2022" it was January 2021 to December 2022? Please be specific.

- "was made per the clinical"?

- "Urinary analysis was performed at baseline" baseline mean free of any treatment?

- "The patients completed a questionnaire" how many times?

- "visual analog scale (VAS) score for pain"  Please tell the readers if the VAS scale is 0 to 5, 0 to 10, 0 to 100 ???

- "The primary endpoint was the change in VAS pain score from baseline to the end of week 4." A piece of information must stay in only one place in the article.

- The primary endpoint was VAS. Please be specific and explain "effect size of 0.6" is this the change in the VAS score? If the VAS pain score is 0 to 10 a change of 0.6 is not clinically relevant.

- Please provide the detection power of the used kits for IL-1β, IL-6, IL-8, IL-10, brain-derived neurotrophic factor 143 (BDNF), tumor necrosis factor alpha (TNF-α), monocyte chemoattractant protein-1 (MCP-144 1), and macrophage inflammatory protein-1 alpha (MIP-1a)  along with the normal ranges.

- "No patient dropped out of the study, and the data of all 28 patients 107 were included in the final analysis" this is a result.

- "4∘C." the "∘" is not the symbol for degree. Please use the appropriate symbol.

- Please provide the ethics approval date.

- It is unclear how the controls were recruited.

Results

- Since VAS is the visual scale it is not correct to report mean and standard deviation.

- List the pain control medications used by the participants.

"All patients were resistant to standard treatments for IC/BPS, including oral medications, intravesical botulinum toxin injection, and hyaluronic acid installation."

- "Among the 28 patients, 4 (all of whom had HIC) had detectable EBV in urine (14.2%, Figure 1), 2 had detectable BKV in urine (7.1%), and 18 had detectable JCV in urine (64.3%)." this is quite different compared to your previously  reported results. Please explain in the Discussion section.

- Please include in Table 1 the baseline characteristics of the control group.

- In Table 2, please let the reader know how many subjects you have in each group. Please pay attention to brackets: (9.25, 20.0) means that 9.25 and 20.0 are not included in the IQR and [9.25, 20.0] means that 9.25 and 20.0 are  included in the IQR.

- A result must stay either in the text or in the table / figure.

- The evaluation of VAS at baseline - week 1- week 2 and week 4 must be described in the Methods section. If such evaluation was done, the test applied is not appropriate.

- Do not use expression such "Conversely," in scientific writing.

- The paired t test is not the appropriate test for any comparisons in Figure 2.

- The use of references are not accepted in this section. "Previous studies have reported higher levels of urinary inflammatory cytokines, including TNF-α, IL-6, IL-8, and IL-10, in patients with IC/BPS.[16,17] Our previous study also demonstrated that levels of urinary inflammatory cytokines are associated with blad-246 der capacity and symptom severity.[18] In addition, intravesical IC/BPS treatment has been reported to be associated with decreased urinary cytokine levels.[19,20]" This information is duplicated and belongs to the Introduction.

- Figures 3 and 4 must be box and whiskers.

- Is the VAS significantly different among men and women?

Discussion

- In the first paragraph of this section, briefly summarizing the main findings.

- And explore possible mechanisms or explanations for your findings.YOu must refer in this process the tables and figures included in the Results section.

- Emphasize the new and important aspects of your study and put your findings in the context of the totality of the relevant evidence.

- State the limitations of your study, and explore the implications of your findings for future research and for clinical practice or policy.

- Discuss the influence or association of confounders on your findings, and the limitations of the data.

- Do not repeat in detail data or other information given in other parts of the manuscript, such as in the Introduction or the Results section.

- "Our findings serve as evidence that EBV infection in bladders with HIC plays a role in the pathophysiology of IC/BPS" this is not supported by your findings.

- You did not appropriately emphasize the limitation of visual VAS in the quantification of the pain.

- You must also appropriately discuss the use of Valacyclovir in the context of the low percentage of viruses and its possible side effects.

Author Response

General comments

- The main concern is related to the applicability; you did not succeed to describe the methods in sufficient detail to allow the reproduction of the study.

Reply: Thanks for your comment. Indeed, the methods in this manuscript were not exhaustively. We added the study protocol and the detail of the procedure for detecting urinary virus in the supplementary files to show the detail of this study (page 3 line 15 to 17; and page 4, line 25 to line 26).

- A change in VAS scale of 0 to 10 equal to 0.6 is without clinical relevance so your study is worthless. How do you quantify a change of 0.6 (the mean is not the appropriate centrality measure for such a small sample)?

Reply: Thanks for your comment. In this study, the results for each group were compared using nonparametric methods, and indeed the data should be presented with median with 25-75 percentile. We had revised the table 1 to show the median and 25-75 percentile of the baseline clinical parameters. The VAS scale change was from 6.78 ± 2.67 [7.5, 5.52-9.0] at baseline to 5.10 ± 3.29 [6, 1.0-7.75] at first week, 4.82 ± 3.16 [6, 0.75-7.0] at second week, and 4.46 ± 2.95 [5, 1.5-6.0] at fourth week; P= 0.0003, 0.0002, and 0.0003, respectively (page 6, line 10, to page 7, line 1). The VAS pain scale was significant in the IC/BPS patients who received valacyclovir treatment. The number of 0.6 was the effect size we used to estimate the patient number in clinical trial.

- You previously reported Epstein-Barr virus 87.5% and 17.4% of patients with HIC and NHIC but your current results are "Of the 28 patients with IC/BPS, 4 had urinary 201 EBV at baseline and none had urinary EBV after treatment." If your hypothesis was that patients had EBV and you treated EBV with valacyclovir 4 subjects, it is a case series, not an original article. Your inclusion criteria were wrong; for such hypothesis, you must include only patients with EBV.  

Reply: Thanks for your comment. In our previous study, EBV in bladder specimens (with in situ hybridization for EBER and PCR) was identified in 87.5% and 17.4% of patients with HIC and NHIC. Both HIC and NHIC patients may had EBV in their bladder specimens. In current study, only 4 patients had urinary EBV, and all of them were HIC. However, EBV undetectable in the urine specimens did not mean that the bladders specimens were free for EBV infection. The IC/BPS patients might still have EBV latency infection in their blader, but their urine samples did not have EBV. Indeed, we should only use valacyclovir to treat IC/BPS patients with EBV, but we could not identify EBV infection only using urine test. Even in IC/BPS patients with bladder EBV infection, the bladder biopsy specimens also may not be EBV positive, because the biopsy may miss the area with EBV infection. Hence, we also included the patients without urinary EBV to evaluate the effect of valacyclovir treatment. We add above discussion into the manuscript (page 11, line 45 to 51)

- You cited up to 10 articles of co-authors out of 30 references. This is too much.

Reply: Thanks for your comment. We had revised the citations remove the self-citations. Currently only 4 references (8, 9, 16 and 18) were published from our research team.

Abstract

- Lines 11-13 incorporate two sentences.

Reply: Thanks for your comment. We had re-wrote the first two sentences in the abstract (page 1, line 11 to 13)

- When and where the study was conducted?

Reply: Thanks for your comment. This study was conducted since 2021 and ended in 2022 in Hualien Tzu Chi Hospital (page 2, line 31 to line 32). Because the words limitation in the abstract is only 200 words, we could not add the information in the abstract section.

- "valacyclovir 500 mg twice a day for 4 weeks" this treatment is current practice or not?

Reply: Thanks for your comment. This treatment currently in not a standard treatment for patients with IC/BPS. For treating EBV infection with valacyclovir, there is no universal dose and duration.

- "urinary inflammatory cytokine level" which cytochine?

Reply: Thanks for your comment. We investigate the level of IL-1β, IL-6, IL-8, IL-10, brain-derived neurotrophic factor (BDNF), tumor necrosis factor alpha (TNF-α), monocyte chemoattractant protein-1 (MCP-1), and macrophage inflammatory protein-1 alpha (MIP-1a) in the urine samples (page 4, line 9 to line 11). Because the words limitation, we could not add the information in the abstract section.

- "The VAS pain score in patients with IC/BPS significantly decreased after 4 weeks of valacyclovir" not specific. Decrease with how many points? Which was the initial VAS and the 4 weeks VAS?

Reply: Thanks for your comment. The VAS pain score in patients with IC/BPS significantly decreased from 7.5 [5.52-9.0] to 5 [1.5-6.0], p=0.0003. We added the numbers to the abstract (page 1, line 21 to 22)

- "the levels of urinary interleukin (IL)-1β, IL-8, IL-10, and tumor necrosis factor-α were decreased." unspecific.

Reply: Thanks for your comment. The levels of urinary interleukin (IL)-1β, IL-8, IL-10, and tumor necrosis factor-α were significantly decreased. We added the word “significantly” to the sentence (page 1, line 24)

- "Valacyclovir could relieve bladder pain, eliminate urinary EBV, and reduce bladder inflammation." this sentence was also true before your study. It is expected to have here the conclusion.

Reply: Thanks for your comment. Our study first evaluated the efficacy of valacyclovir in treating the patients with IC/BPS.

Introduction

- Please avoid repetition of "bladder" in the same sentence.

Reply: Thanks for your comment. We had removed some redundant word “bladder” (page 1, line 31; and page 2, line 2).

- Please do not use expressions such as "A recent study".

Reply: Thanks for your comment. We had removed the word “A recent study” (page 1, line 31).

- "Previous studies also revealed upregulation of inflammatory cytokines, such as interleukin-6 (IL-6), IL-17, and tumor necrosis factor-α (TNF-α) in the bladders of IC/BPS.[6,7]" Please be specific.

Reply: Thanks for your comment. We had revised the above sentence. Previous studies had used immunochemical staining, polymerase chain reaction (PCR) and Next-Generation RNA sequencing to show upregulation of inflammatory cytokines, such as interleukin-6 (IL-6), IL-17, and tumor necrosis factor-α (TNF-α) in the bladders of IC/BPS. (page 1, line 35 to 38)

- "The current therapeutic strategy for IC/BPS primarily involves reducing bladder inflammation and protecting the urothelium, such as intravesical botulinum toxin injection or hyaluronic acid installation [1]."  Which is the efficacy?

Reply: Thanks for your comment. We had revised the above sentence. The current therapeutic strategy for IC/BPS primarily involves reducing bladder inflammation and protecting the urothelium, such as intravesical botulinum toxin injection or hyaluronic acid installation, and studies revealed significant bladder pain and urinary tract symptoms improvement after the treatments. However, symptoms relapse is common in patients with IC/BPS and currently the curative treatment for IC/BPS is not available. (page 1, line 38 to 43)

- "Our previous study was the first to provide evidence that Epstein-Barr virus (EBV) infection was present in the bladder of patients with IC/BPS" how frequent?

Reply:

Reply: Thanks for your comment. EBV infection was identified in the bladder specimens of 87.5% and 17.4% of patients with HIC and NHIC, respectively. (page 2, line 6 to 7)

- "In situ" should be read as "in situ".

Reply: Thanks for your comment. We had revised it. (page 2, line 5)

- Write the aim of your study at past tense.

Reply: Thanks for your comment. We had revised the sentence. (page 2, line 26)

Materials and Methods

- "2021 and 2022" it was January 2021 to December 2022? Please be specific.

Reply: Thanks for your comment. Patients with IC/BPS were prospectively enrolled between 2021 February and 2022 April in Hualien Tzu Chi Hospital. (page 2, line 31 to 32)

- "was made per the clinical"?

Reply: Thanks for your comment. It should be made according to. We had revised the sentence: “The diagnosis of IC/BPS was made according to the clinical symptom in American Urology Association guidelines”. (page 2, line 32 to line 33)

- "Urinary analysis was performed at baseline" baseline mean free of any treatment?

Reply: Thanks for your comment. It should be before the valacyclovir treatment. All of the patients had previously received the others treatment for IC/BPS. We had revised the sentence. (page 2, line 39 to 40)

- "The patients completed a questionnaire" how many times?

Reply: Thanks for your comment. The patients also completed the questionnaires including VAS pain scale, ICPI, ICSI, QoL at the end of weeks 1, 2, and 4 after treatment. (page 3, line 7 to 9)

- "visual analog scale (VAS) score for pain" lease tell the readers if the VAS scale is 0 to 5, 0 to 10, 0 to 100 ???

Reply: Thanks for your comment. The visual analog scale (VAS) score for pain was ranged from 0 to 10. (page 2, line 44 to 45)

- "The primary endpoint was the change in VAS pain score from baseline to the end of week 4." A piece of information must stay in only one place in the article.

Reply: Thanks for your comment. This information only showed once in the main text of this article. (page 3, line 10 to 11)

- The primary endpoint was VAS. Please be specific and explain "effect size of 0.6" is this the change in the VAS score? If the VAS pain score is 0 to 10 a change of 0.6 is not clinically relevant.

Reply: Thanks for your comment. The effect size is a value measuring the strength of the relationship between two variables in a population. It is widely used in statistical hypothesis testing, sample size planning and meta-analyses studies. In medical researches, effect sizes usually are considered small if they are less than 0.2, moderate if they are between 0.5 and 0.8, and large if they are greater than 0.8 (Med Care. 1989 Mar;27(3 Suppl):S178-89). In this study, we suppose the valacyclovir treatment of effect was moderate and used effect size 0.6 to estimate sample size in this study. We added above explanation in the method section. (page 3, line 19 to line 20)

- Please provide the detection power of the used kits for IL-1β, IL-6, IL-8, IL-10, brain-derived neurotrophic factor 143 (BDNF), tumor necrosis factor alpha (TNF-α), monocyte chemoattractant protein-1 (MCP-144 1), and macrophage inflammatory protein-1 alpha (MIP-1a) along with the normal ranges.

Reply: Thanks for your comment. The detection powers of the biomarkers were provided by the manufacture: BDNF: 2.4-10,000 pg/mL, IL-1β: 0.5 - 25,000 pg/mL, IL-6:0.24 - 10,000 pg/mL, IL-8: 0.14 - 10,000 pg/mL, L-10: 0.7 - 40,000 pg/mL, TNF-α: 1.2 - 100,000 pg/ml, MCP-1: 3 - 50,000 pg/mL, MIP-1α: 0.3 - 50,000/mL. All data of the biomarkers presented in this study was in the ranges between the detection power. We added the detection power into the supplementary material. Currently, there was no standard normal ranges for this biomarker in human urine samples.

- "No patient dropped out of the study, and the data of all 28 patients 107 were included in the final analysis" this is a result.

Reply: Thanks for your comment. We had deleted this sentence in the method section and added into the result section. (page 3, line 24 to line 25; and page 4, line 42 to 43)

- "4∘C." the "∘" is not the symbol for degree. Please use the appropriate symbol.

Reply: Thanks for your correction. We had corrected the errors in the manuscript. (from page 3 ,line 29 to page 4, line 15)

- Please provide the ethics approval date.

Reply: Thanks for your comment. The IRB for this study was approved on July 1st, 2019. We added this information to the supplementary protocol.  

- It is unclear how the controls were recruited.

Reply: Thanks for your comment. Patients who visited the urology clinic for anti-incontinence surgery were asked to provide a urine sample as a control for urinary virus investigations. The control patients also had received urine analysis test to exclude asymptomatic urinary tract infection and received urodynamic study to exclude possible bladder diseases such as overactive bladder. (page 3, line 11 to 15)

Results

- Since VAS is the visual scale it is not correct to report mean and standard deviation.

Reply: Thanks for your comment. We added the median and the 25th-75th percentile into the result section and the table 1. (page 4, line 46 to 47)

- List the pain control medications used by the participants.

Reply: Thanks for your comment. The patients used acetaminophen or celecoxib for pain control. (page 4, line 45)

"All patients were resistant to standard treatments for IC/BPS, including oral medications, intravesical botulinum toxin injection, and hyaluronic acid installation."

Reply: Thanks for your comment. It should be refractory. We had revised it. (page, line )

- "Among the 28 patients, 4 (all of whom had HIC) had detectable EBV in urine (14.2%, Figure 1), 2 had detectable BKV in urine (7.1%), and 18 had detectable JCV in urine (64.3%)." this is quite different compared to your previously reported results. Please explain in the Discussion section.

Reply: Thanks for your comment. In our previous study, we reported higher proportion of detectable EBV in the IC/BPS bladders, and current study showed the relatively lower proportion of detectable EBV in the urine samples from patients with IC/BPS. IC/BPS patients with EBV in their bladders may not also have detectable urinary EBV. (page 11, line 45 to 48)

- Please include in Table 1 the baseline characteristics of the control group.

Reply: Thanks for your comment. We had revised the table 1. (page 5, line 10 to 15)

- In Table 2, please let the reader know how many subjects you have in each group. Please pay attention to brackets: (9.25, 20.0) means that 9.25 and 20.0 are not included in the IQR and [9.25, 20.0] means that 9.25 and 20.0 are included in the IQR.

Reply: Thanks for your comment. We had added the patient numbers in each group. We also revised the brackets in the table 2. (page 5, line 16 to 20)

- A result must stay either in the text or in the table / figure.

Reply: Thanks for your comment. We had removed some data from method section to result section.

- The evaluation of VAS at baseline - week 1- week 2 and week 4 must be described in the Methods section. If such evaluation was done, the test applied is not appropriate.

Reply: Thanks for your comment. We had added the timing of evaluation VAS in the method section. (page 3, line 7 to line 9)

- Do not use expression such "Conversely," in scientific writing.

Reply: Thanks for your comment. We had removed the word. (page 7, line 3)

- The paired t test is not the appropriate test for any comparisons in Figure 2.

Reply: Thanks for your comment. It should be Wilcoxon matched-pair signed-rank test. We had revised the legend of Figure 2. (page 8, line 1)

- The use of references are not accepted in this section. "Previous studies have reported higher levels of urinary inflammatory cytokines, including TNF-α, IL-6, IL-8, and IL-10, in patients with IC/BPS.[16,17] Our previous study also demonstrated that levels of urinary inflammatory cytokines are associated with blad-246 der capacity and symptom severity.[18] In addition, intravesical IC/BPS treatment has been reported to be associated with decreased urinary cytokine levels.[19,20]" This information is duplicated and belongs to the Introduction.

Reply: Thanks for your comment. We had removed above mentioned sentence. (page 8, line 22 to line 26)

- Figures 3 and 4 must be box and whiskers.

Reply: Thanks for your comment. We had revised the Figure 3 and Figure 4 and used box and whiskers to present the data.

- Is the VAS significantly different among men and women?

Reply: Thanks for your comment. There were only 4 male patients with IC/BPS in this study, and the VAS pain scale was not significantly different between men and women.

Discussion

- In the first paragraph of this section, briefly summarizing the main findings.

Reply: Thanks for your comment. The summary of this study showed in the end of the first paragraph of discussion section. (page 11, line 6 to line 13)

- And explore possible mechanisms or explanations for your findings. You must refer in this process the tables and figures included in the Results section.

Reply: Thanks for your comment. We added the refers to the figures in discussion section. (page 11, line 25, line 27, line 46, line 53 and page 12, line 1, line 5, line 10 and line 11)

- Emphasize the new and important aspects of your study and put your findings in the context of the totality of the relevant evidence.

Reply: Thanks for your comment. We described the novel finding in this study and the relevant evidence. (page 11, line 24 to line 29)

- State the limitations of your study, and explore the implications of your findings for future research and for clinical practice or policy.

Reply: Thanks for your comment. The we added the limitations in the last paragraph of discussion section. (page 12, line 25 to line 29)

- Discuss the influence or association of confounders on your findings, and the limitations of the data.

Reply: Thanks for your comment. The we added the limitations in the last paragraph of discussion section. (page 12, line 1 to line 4; and page 12, line 17 to line 29)

- Do not repeat in detail data or other information given in other parts of the manuscript, such as in the Introduction or the Results section.

Reply: Thanks for your comment. We had deleted some repeated information in the discussion section. (page 11, line 14 to 17 and page 11, line 40 to 41)

- "Our findings serve as evidence that EBV infection in bladders with HIC plays a role in the pathophysiology of IC/BPS" this is not supported by your findings.

Reply: Thanks for your comment. It should be the conclusion of our previous published study. We added the reference to the sentence. (page 12, line 35 to 36)

- You did not appropriately emphasize the limitation of visual VAS in the quantification of the pain.

Reply: Thanks for your comment. Indeed, using the VAS pain scale of the pain may not accurately method to quantify pain in the patients, because they needed to select from a finite number of pain description and may not find a number that appropriately reflects their discomfort. (page 12, line 29 to 32)

- You must also appropriately discuss the use of Valacyclovir in the context of the low percentage of viruses and its possible side effects.

Reply: Thanks for your comment. Oral valacyclovir might cause adverse effect such as nausea, vomiting, headaches, dizziness, diarrhea, or constipation. For the IC/BPS patients without urinary EBV, clinicians should consider the benefit and potential adverse effect of the valacyclovir treatment. (page 12, line 21 to 24)

Round 2

Reviewer 3 Report

Comments and Suggestions for Authors

Thank you for your effort to improve your manuscript.

Abstract

- Please provide centrality and dispersion metrics of interleukin (IL)-1β, IL-8, IL-10, and tumor necrosis factor-α

Methods

- "used effect size of 0.6" it is still unclear from where you take this value considering that mean is not an appropriate cen\rrtality metric for VAS.

- I do not see why details of methods are provided in the supplementary file. This practice does not necessarily support reproductivity.

- No information on when and where the study was conducted is available in this section.

- The study recruited patients (February 2021) before registering the study protocol (2021-10-26). This is a matter of research integrity.

Results

- It is expected to have also men in the control group.

- A specific result is presented more than once in this section (e.g., "Of the 28 patients with IC/BPS, 4 had urinary EBV at baseline and none had urinary EBV after treatment").

Discussion

- Do not start this section with reference to teh scientific literature.

Author Response

Abstract

- Please provide centrality and dispersion metrics of interleukin (IL)-1β, IL-8, IL-10, and tumor necrosis factor-α

Reply: Thanks for your comments and suggestion. The centrality and dispersion metrics of IL-1β, IL-8, IL-10, and TNF-α were indeed important, however, the word limitation of abstract is only 200 words in this journal. We added the centrality and dispersion metrics of interleukin (IL)-1β, IL-8, IL-10, and tumor necrosis factor-α in the abstract section. (the decreases of urinary interleukin (IL)-1β (from 0.66 [0.55-0.82] pg/ml to 0.58 [0.55-0.64] pg/ml, p=0.0034), IL-8 (from 6.81 [2.38 to 29.1] pg/ml to 4.33 [1.53-11.04] pg/ml, p=0.0361), IL-10 (from 1.06 [0.94-1.18] pg/ml to 0.92 [0.88-1.02], p=0.0086), and tumor necrosis factor-α (from 1.61 [1.50-1.72] pg/ml to 1.50 [1.44-1.55] pg/ml, p=0.0079) were significant.) (page 1, line 23 to line 27) The others detail data were showed in the result section Figure 3.

Methods

- "used effect size of 0.6" it is still unclear from where you take this value considering that mean is not an appropriate cenrrtality metric for VAS.

Reply: Thanks for your comments. According to previous studies, usually the effect size of VAS pain scale in treatment for IC/BPS were medium (Toxins (Basel). 2019 Sep; 11(9): 510.), and most studies of IC/BPS used VAS pain scales to evaluate treatment outcome. We added above mentioned reference in the method section. (page 3, line 29 to line 30)

- I do not see why details of methods are provided in the supplementary file. This practice does not necessarily support reproductivity.

Reply: Thanks for your comments. We added some information of the clinical trial in the method section to increase the reproductivity. (page 2, line 37 to line 41; page 2, line 46; page 2, line 49 to line 50; page 3, line 18 to line 19)

- No information on when and where the study was conducted is available in this section.

Reply: Thanks for your comments. This study was started in November 2021 and ended in August 2022 (from patient enrollment to laboratory investigations), and was conducted in Hualien Tzu Chi Hospital, Hualien, Taiwan. We added above information in the method section. (page 2, line 33 to line 35)

- The study recruited patients (February 2021) before registering the study protocol (2021-10-26). This is a matter of research integrity.

Reply: Thanks for your correction. We had checked our records and found it is a mistake. We recruited the patients since November 2021 and ended in April 2022. We had revised the date of patient enrollment (page 2, line 35 to 37) Thanks again to help us to find the mistake.

Results

- It is expected to have also men in the control group.

Reply: Thanks for your comments. Indeed, men also should be included in the control group although most patients with IC/BPS were women. The sexual difference might cause urinary viral spectrum difference. However, we currently only had the urine samples from female control subjects. We added above bias into the limitation section. (page 13, line 23 to 25)

- A specific result is presented more than once in this section (e.g., "Of the 28 patients with IC/BPS, 4 had urinary EBV at baseline and none had urinary EBV after treatment").

 Reply: Thanks for your comments. Indeed the result was repeatedly. We had deleted the sentence in the result section. (page 6, line 7 to line 8)

Discussion

- Do not start this section with reference to the scientific literature.

Reply: Thanks for your suggestion. We removed the reference to the scientific literature in the beginning of this section and started this section with the summary of this study. (page 12, line 2 to line 6)